# Taming Cross-Domain Representation Variance in Federated Prototype Learning with Heterogeneous Data Domains

**Lei Wang** [*]
University of Florida
Gainesville, FL 32611
leiwang1@ufl.edu

**Jieming Bian** [*]
University of Florida
Gainesville, FL 32611
jieming.bian@ufl.edu

**Letian Zhang**
Middle Tennessee State University
Murfreesboro, TN 37132
letian.zhang@mtsu.edu

**Chen Chen**
University of Central Florida
Orlando, FL 32816
chen.chen@crcv.ucf.edu

**Jie Xu**
University of Florida
Gainesville, FL 32611
jie.xu@ufl.edu

## Abstract

Federated learning (FL) allows collaborative machine learning training without sharing private data. While most FL methods assume identical data domains across clients, real-world scenarios often involve heterogeneous data domains. Federated Prototype Learning (FedPL) addresses this issue, using mean feature vectors as prototypes to enhance model generalization. However, existing FedPL methods create the same number of prototypes for each client, leading to cross-domain performance gaps and disparities for clients with varied data distributions. To mitigate cross-domain feature representation variance, we introduce FedPLVM, which establishes variance-aware dual-level prototypes clustering and employs a novel $\alpha$-sparsity prototype loss. The dual-level prototypes clustering strategy creates local clustered prototypes based on private data features, then performs global prototypes clustering to reduce communication complexity and preserve local data privacy. The $\alpha$-sparsity prototype loss aligns samples from underrepresented domains, enhancing intra-class similarity and reducing inter-class similarity. Evaluations on Digit-5, Office-10, and DomainNet datasets demonstrate our method's superiority over existing approaches.

## 1 Introduction

Federated Learning [21] (FL) is a novel distributed learning framework that enables clients to collaboratively train a global model using their respective local datasets, thereby preserving data privacy. FL offers distinct advantages over traditional distributed learning methodologies by mitigating communication costs and addressing privacy concerns, which has led to its increased adoption across various sectors. Despite these benefits, FL comes with its own challenges, especially in terms of data heterogeneity [41]. In FL, clients gather private data from unique sources, resulting in non-independent and identically distributed (non-IID) datasets. Such non-IID distributions can lead clients toward their local optima, potentially diverging from the global objective. Consequently, this may impede convergence rates and diminish overall model performance [17].

To address the aforementioned issue of data heterogeneity, numerous FL techniques have been devised [20, 5, 41, 29, 14, 12]. While these methods have indeed improved convergence, their focus on non-IID scenarios remains largely restricted. Typically, they assume that the data across clients

---

[*]The first two authors contributed equally to this work.

38th Conference on Neural Information Processing Systems (NeurIPS 2024).

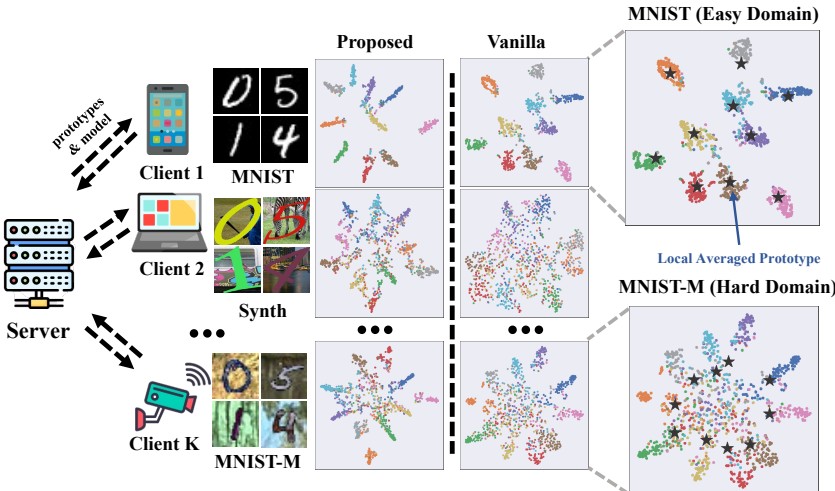

Figure 1: **Illustration of federated learning with heterogeneous data domains.** The **Vanilla** column depicts the local feature distribution of the standard FedPL approach, obtaining average local and global prototypes directly. **Proposed** method showcased in the adjacent column yields a larger inter-class distance and a reduced intra-class distance. **Note that** without capturing variance information, even for hard domains, local averaged prototypes for each class can be well distinguished while the feature vectors are still mixed up. Both methods illustrate noticeable variations in domain characteristics across datasets, as detailed in Fig. 4.

pertain to a single domain, attributing non-IID distribution to label skew alone. Yet, in more realistic scenarios, clients gather data according to their unique preferences, making it impractical to assume identical domain origins for local data. *Instead, the data often stems from heterogeneous domains, resulting in varied feature distributions.*

Current approaches [10, 32] to FL in heterogeneous domains aim to develop a global prototype for each label category. These prototypes are designed to minimize the distance between the individual training samples and their corresponding category's global prototype. Typically, a global prototype is computed as the mean of the local prototypes from each client, whereas a local prototype is itself the average of the representations of samples within the same category. Several studies have proposed unique designs to enhance training performance in these settings. For example, one work [33] enables contrastive learning at the client level by facilitating the exchange of local prototypes, thus promoting inter-client knowledge transfer. However, this exchange raises privacy concerns and substantially increases communication overhead. Another study [10] opts for a clustering approach to identify representative prototypes, thereby preserving domain diversity and preventing bias towards predominant domains. This technique has proven effective, especially in scenarios with disproportionate client distribution across domains.

Although these methods improve overall performance across domains, they do not address the unequal learning challenges that arise from domain diversity. For instance, on the Digits dataset [42], methods may perform well in domains like MNIST [7] but underperform in more challenging domains such as SVHN [24]. This discrepancy is evident in representation distributions, as exemplified in Fig. 1, where 'easy' domains show tight clustering of samples within the same category and clear separation between different categories, facilitating accurate classification. In contrast, 'hard' domains display looser clustering, increasing the likelihood of misclassification, particularly for samples near category boundaries. Addressing these disparities is crucial for the equitable advancement of FL methodologies across diverse client domains.

Considering the challenges posed by unequal learning due to domain diversity, we introduce a novel method, termed FedPLVM (short for Federated Prototype Learning with Variance Mitigation). FedPLVM devises two main mechanisms. **Firstly**, we develop a dual-level prototype clustering mechanism that adeptly captures variance information, a significant improvement over previous methodologies that rely on averaging local training samples' representations to derive local prototypes. Our local-level clustering generates multiple local clustered prototypes within each domain. To further mitigate increased communication costs and privacy concerns arising from transferring a

comprehensive set of diverse prototypes for each class from every client, we employ global-level prototype clustering on the server side. **Secondly**, by capturing the variance information through clustered prototypes, we design an innovative $\alpha$-sparsity prototype loss to enhance the training process. To prevent the intermingling of feature distributions from 'hard' clients, instead of simply maximizing the feature-level distance between each local instance and the prototypes of other classes, we refine this distance by elevating it to an $\alpha$ power, with $\alpha$ being a value between 0 and 1. This modification effectively repels other prototypes, thereby introducing greater sparsity in the inter-class feature distributions. Moreover, we incorporate an intra-class similarity correction term to lessen the feature-level distance among intra-class samples, thereby concentrating the $\alpha$-sparsity prototype loss on amplifying the feature-level distance across inter-class samples. Collectively, these two strategies empower FedPLVM to reasonably leverage variance information, thus equilibrating fairness between 'hard' and 'easy' learning domains and enhancing overall learning performance. Consequently, FedPLVM emerges as a reliable approach for Federated Learning in contexts characterized by heterogeneous data domains. Our main contributions are outlined as follows:

- This study delves into FL with heterogeneous data domains, examining why models exhibit varying performance across different domains. We identify a fundamental limitation in existing methods: their inability to effectively address the disparate learning challenges inherent in diverse domains.

- To tackle these uneven learning challenges, we introduce a novel approach, FedPLVM. This method incorporates a dual-level prototype clustering method, capturing the rich sample representation variance information while ensuring communication efficiency. Additionally, we develop a new $\alpha$-sparsity prototype loss to address learning difficulties more equitably.

- Extensive experiments conducted on the Digit-5 [42], Office-10 [8], and DomainNet [26] datasets demonstrate the superior performance of our proposed method when compared with multiple state-of-the-art approaches.

## 2 Related Work

**Federated Learning.** FL aims to train a global model through collaboration among multiple clients while preserving their data privacy. FedAvg [21], the pioneering work in FL, demonstrates the advantages of this approach in terms of privacy and communication efficiency by aggregating local model parameters to train a global model. A significant challenge in FL is data heterogeneity, often manifested as non-IID (independently and identically distributed) data. Subsequent research, following FedAvg, has primarily focused on addressing data heterogeneity to enhance training performance in FL environments. Specifically, studies such as [18, 2, 30] have improved performance by incorporating a global penalty term to mitigate discrepancies. Other works, e.g., [32, 23], have sought to maximize feature-level agreement between local and global models to further boost performance. Recent works [10, 32, 19, 40] have explored data heterogeneity arising from client-specific domain diversity, while overlooking the challenges of unequal learning across different domains. In this paper, we propose a novel approach using dual-level prototype clustering to capture essential local variance information. Additionally, we introduce a new $\alpha$-sparsity loss, specifically designed to tackle the challenges of learning in diverse domains, thereby facilitating the development of a more generalizable global model.

**Prototype Learning.** Prototype learning has been extensively explored in various tasks, such as transfer learning [27, 13], few-shot learning [31], zero-shot learning [11], and unsupervised learning [34]. The concept of a prototype in this context refers to the average feature vectors of samples within the same class. In the FL literature, prototypes serve to abstract knowledge while preserving privacy. Specifically, approaches like FedProc [23] and FedProto [32] focus on achieving feature-wise alignment with global prototypes. FedPCL [33] employs prototypes to capture knowledge across clients, constructing client representations in a prototype-wise contrastive manner using a set of pre-trained models. FPL [10] highlights the use of cluster-based and unbiased global prototypes to tackle the challenges in FL where clients possess domain-diverse data. Our study addresses a similar issue as FPL but with distinct emphases. While FPL concentrates on the disparities in the number of clients across domains, aiming to mitigate the bias in the overall model caused by domains with more clients, our work focuses on the intrinsic learning challenges that vary across domains.

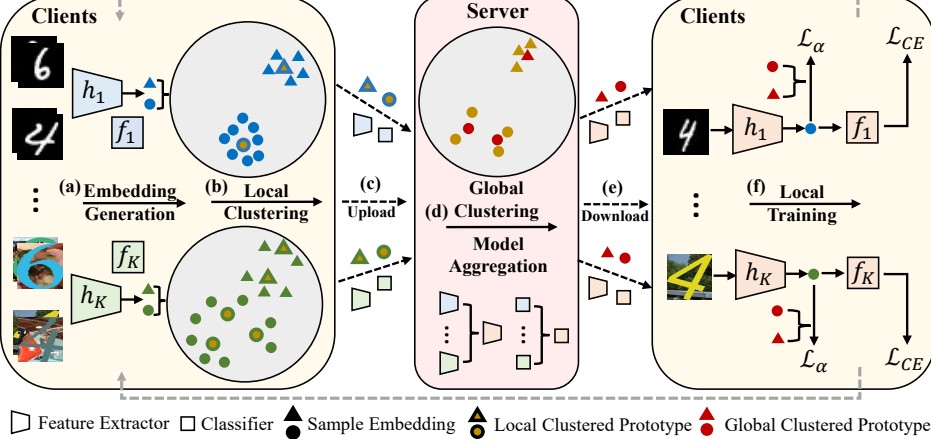

Figure 2: **An overview of our proposed FedPLVM framework. Once** the sample embedding is generated by the feature extractor, the client conducts the first-level local clustering, following Eq. 3. **Subsequently**, the server gathers all local clustered prototypes and local models (comprising feature extractors and classifiers), initiates the second-level global clustering based on Eq. 4, and averages the local models to form a global model. **Finally**, clients utilize the received global clustered prototypes to update the local model, employing loss functions $\mathcal{L}_\alpha$ from Eq. 5 and $\mathcal{L}_{CE}$ from Eq. 9.

**Contrastive Learning.** Contrastive learning is a promising self-supervised learning technique. The work by [25] constructs pairs of positives and negatives for each sample and applies the InfoNCE loss to compare these pairs. Another work by [15] extends contrastive learning from self-supervised to fully supervised settings, utilizing both label information and contrastive methods. Additionally, some studies [16, 39] integrate contrastive learning into local training to enhance performance in FL. Instead of the conventional InfoNCE loss, our approach introduces a new $\alpha$-sparsity loss, aiming to further reduce similarity among inter-class sample features while amplifying similarity among intra-class samples.

## 3 Preliminary

**Regular FL Scenario.** Consider the classic FL scenario: there exist $K$ clients and one server, which aims to assist all clients to train a common machine learning model without sharing their private data, denoted by $\mathcal{D}_k = \{\mathbf{x}_i, y_i\}_i^{N_k}$ for client $k$. Formally, the global objective of FL can be formulated as:

$$\min_w \sum_{k=1}^K \frac{N_k}{N} \mathcal{L}_k(w; \mathcal{D}_k), \tag{1}$$

where $L_k$ is the local loss function for client $k$, $w$ and $N = \sum_{k=1}^K N_k$ denote the shareable model and the total number of samples among all clients respectively.

**Domain Shift in FL.** In the simplest FL setting, the label and feature distribution are the same between two clients $i$ and $j$, formally $P_i(y) = P_j(y)$ and $P_i(x|y) = P_j(x|y)$. Extending such a scenario to the heterogeneous data setting brings us to the point of domain shift, also denoted as feature non-IID setting. Domain shift is caused by distinctive feature distributions among clients, that is $P_i(x|y) \neq P_j(x|y)$, though their label space is still the same.

**Federated Prototype Learning.** To handle the FL domain shift problem, previous works [32, 10] divide the classification network into two parts: feature extractor and classifier. The feature extractor $h : \mathbb{R}^V \to \mathbb{R}^D$ maps a sample $\mathbf{x} \in \mathbb{R}^V$ into the feature space and generates the corresponding feature vector $\mathbf{z} = h(\mathbf{x}) \in \mathbb{R}^D$. Then the classifier $f : \mathbb{R}^D \to \mathbb{R}^M$ outputs the $M$-class prediction $f(\mathbf{z}) = f(h(\mathbf{x})) \in \mathbb{R}^M$ given the feature vector $\mathbf{z}$. The intuition of FedPL is adjusting the feature extractor to generate a consistent feature distribution among different domains. Similar to the global shareable model, the straightforward solution computes the average feature vector of all samples belonging to the same class for further processing, called the **prototype**. Due to the distributed property of FL, we cannot collect all samples via the server, which makes the above-mentioned idea

a two-step procedure:

$$\bar{p}_k^m = \frac{1}{|\mathcal{D}_k^m|} \sum_{(\mathbf{x}_i, y_i) \in \mathcal{D}_k^m} h(\mathbf{x}_i), \quad \bar{g}^m = \frac{1}{K} \sum_{k=1}^{K} \bar{p}_k^m, \quad \forall k \in \mathcal{K}, m \in \mathcal{M}, \tag{2}$$

where $\mathcal{K} = \{1, 2, \ldots, K\}$ is the set of clients, $\mathcal{M} = \{1, 2, \ldots, M\}$ is the set of classes. $\bar{p}_k^m$ and $\bar{g}^m$ denotes the **local prototype** of class $m$ on client $k$ and the **global prototype** of class $m$, respectively. $\mathcal{D}_k^m$ is the subset that all samples belonging to class $m$ in the local dataset $\mathcal{D}_k$. Upon obtaining the prototypes, most FedPL works tend to design a loss function that approximates the prototype of one's own class while staying away from the prototypes of others.

## 4 FedPLVM: FedPL with Variance Mitigation

### 4.1 Dual-Level Prototype Generation

**Local Prototype Clustering.** To mitigate the impact of domain variance in FedPL, we revise the two-step averaged prototype generation process to a dual-level clustering algorithm. In Fig. 1, the pronounced diversity in domain variances among clients becomes evident, intricately linked to the complexity of their datasets. For instance, comparing the feature distribution between Synth and MNIST shows a visibly more scattered pattern in Synth due to its higher complexity, whereas MNIST, being comparatively easier to learn, displays a more structured distribution. It becomes evident that computing a single local averaged prototype for one class per client is 'unfair', given the differing richness of feature distribution information among clients. Particularly, for 'hard' clients with complex datasets such as SVHN, employing multiple local prototypes becomes imperative to capture the scattered feature distribution comprehensively. Hence, we propose the first-level prototype generation, namely local prototype clustering. Instead of a straightforward averaging approach, our method involves initially clustering the feature vectors of all same-class local samples, forming several local clustered prototypes as a set of local representations.

$$\mathcal{P}_k^m = \left\{ p_{k,j}^m \right\}_{j=1}^{J_{k,m}} \xleftarrow{Cluster} \left\{ h(\mathbf{x}_i) | (\mathbf{x}_i, y_i) \in \mathcal{D}_k^m \right\}, \tag{3}$$

where $J_{k,m}$ and $p_{k,j}^m$ represents the number of local clustered prototypes and the $j$-th local clustered prototype of class $m$ on client $k$ clustered from the set of feature vectors $\{h(\mathbf{x}_i) | (\mathbf{x}_i, y_i) \in \mathcal{D}_k^m\}$. It is important to note that the number of local clustered prototypes may differ across various classes and clients. To determine these prototypes, we employ the parameter-free clustering algorithm, FINCH [28], utilizing cosine similarity as the clustering metric. We also conduct additional experiments on comparison with other clustering methods in Supplementary Material. This choice ensures the alignment of the number of local clustered prototypes with the sparsity of the domain distribution. By leveraging this approach, we enhance the representation of distinct feature distributions, preventing the overdrift of local averaged prototypes towards densely concentrated regions in the feature space.

**Global Prototype Clustering on Server.** Distributing all local clustered prototypes among clients poses challenges due to the extra communication cost and privacy concerns. Hence, we introduce the second level of prototype generation, namely global prototype clustering, which can be formulated as:

$$\mathcal{G}^m = \left\{ g_j^m \right\}_{j=1}^{C_m} \xleftarrow{Cluster} \mathcal{P}^m = \left\{ \mathcal{P}_k^m \right\}_{k=1}^{K}, \tag{4}$$

where $\mathcal{C}_m$ and $g_j^m$ denote the number of global clustered prototypes and the $j$-th global clustered prototype of class $m$ on the server from the local collected prototypes set $\mathcal{P}^m = \{\mathcal{P}_k^m\}_{k=1}^{K}$. Through the second-level global clustering, we significantly reduce the number of prototypes that the server must distribute to clients compared to distributing all locally clustered prototypes, alleviating potential communication costs. Dual-level clustering also addresses privacy concerns that arise from original local clustered prototypes potentially revealing client-specific features and the corresponding results can be found in Supplementary Material.

### 4.2 $\alpha$-Sparsity Prototype Loss

Unlike previous methods in FedPL, which generate a single prototype per class, our approach employs dual-level clustering to create multiple prototypes for each class, thereby capturing valuable

variance information. This multiplicity of prototypes could potentially lead to overlapping feature representations among different classes, especially in challenging client scenarios. To mitigate this risk, we introduce a novel $\alpha$-sparsity prototype loss, inspired by the InfoNCE-based loss. Our newly designed $\alpha$-sparsity prototype loss enhances inter-class feature distribution sparsity and maintains balanced feature representation distances within classes, unlike the traditional InfoNCE-based loss. The formulation of the $\alpha$-sparsity prototype loss is detailed below:

$$\mathcal{L}_{\alpha} = \mathcal{L}_{contra} + \mathcal{L}_{corr}. \tag{5}$$

The first contrastive term can be formulated as:

$$\mathcal{L}_{contra} = -\log \frac{\sum\limits_{g^{y_i} \in \mathcal{G}^{y_i}} \exp\left(s_{\alpha}(h(\mathbf{x}_i), g^{y_i})/\tau\right)}{\sum\limits_{g \in \mathcal{G}} \exp\left(s_{\alpha}(h(\mathbf{x}_i), g)/\tau\right)}, \tag{6}$$

where $\tau$ is the temperature hyper-parameter that controls the concentration strength of the similarity [35], and $\mathcal{G} = \{\mathcal{G}^m\}_{m=1}^M$ is the set of all global clustered prototypes. $s_{\alpha}(\cdot, \cdot)$ in the first term is the modified $\alpha$-sparsity cosine similarity metric between the feature vector $h(\mathbf{x}_i)$ and the prototype $g^m$ from class $m$ and can be formulated as:

$$s_{\alpha}(h(\mathbf{x}_i), g^m) = \left(\frac{h(\mathbf{x}_i)}{||h(\mathbf{x}_i)||} \cdot \frac{g^m}{||g^m||}\right)^{\alpha}, \tag{7}$$

where $\alpha \in (0, 1)$. This metric serves to compel the feature extractor to generate feature outputs closely aligned with the global clustered prototypes of their respective classes while distancing them from other global prototypes. It achieves this by maximizing intra-class similarity and minimizing inter-class similarity. As our feature vectors have positive values, cosine similarity always falls within the range of $[0, 1]$. By introducing a sparsity factor $\alpha$ to the similarity and applying it as a power, all similarity values are elevated. However, due to the smaller denominator component representing similarity with other classes in the $\mathcal{L}_{\alpha}$ function, the impact of the concave function $(\cdot)^{\alpha}$ is more pronounced. This emphasis on maximizing inter-class distance is what we denote as $\alpha$-**sparsity** operation, which directs more attention toward expanding the overall feature distribution to a broader range. Consequently, it mitigates the issue of feature distributions within one class being excessively dispersed and overlapping with feature distributions of other classes.

This modification does indeed lead to an unintended consequence: an increase in intra-class distance due to heightened similarity between the feature vector and prototypes of the corresponding class, resulting from the concavity of $(\cdot)^{\alpha}$. This brings us to the second correction term of our $\alpha$-sparsity prototype loss:

$$\mathcal{L}_{corr} = \left|\left| \sum_{g^{y_i} \in \mathcal{G}^{y_i}} s_{\alpha}(h(\mathbf{x}_i), g^{y_i}) - C_{y_i} \right|\right|_2. \tag{8}$$

The latter term within the $\alpha$-sparsity prototype loss serves as a corrective measure, which focuses on pushing the average cosine similarity closer to 1. This correction measure aims to counterbalance the increase in intra-class distance stemming from the adjustment introduced by $\alpha$.

An additional Cross-Entropy (CE) loss [6] is employed to train the classifier and derive prediction results, which can be formulated as:

$$\mathcal{L}_{CE} = \sum_{(\mathbf{x}_i, y_i) \in \mathcal{D}_k} -\mathbf{1}_{y_i} \log(f(h(\mathbf{x}_i))). \tag{9}$$

The total local loss, combining the previously mentioned loss functions, is expressed as follows:

$$\mathcal{L}_{local} = \lambda \mathcal{L}_{\alpha} + \mathcal{L}_{CE}. \tag{10}$$

Here, $\lambda$ serves as a hyper-parameter that regulates the balance between the $\alpha$-sparsity prototype loss and the CE loss. This formulation allows for a unified and weighted consideration of the $\alpha$-sparsity prototype loss.

In summary, our FedPLVM operates as follows in each training round: Initially, each client generates feature vectors for all local samples and clusters these into several local clustered prototypes. These local clustered prototypes are then uploaded to the server, which aggregates them into distinct

prototype sets for various classes and further clusters them to form global clustered prototypes. Concurrently, the server collects local models from all clients and consolidates them into a unified global model. The server then distributes all global clustered prototypes and global model to clients. Subsequently, each client utilizes the global clustered prototypes to train the global model on its private dataset using $\mathcal{L}_{local}$, obtaining its local model. Finally, clients employ their local models to generate new feature vectors and repeat the aforementioned procedure in the next training round. For a detailed insight into FedPLVM, refer to Algorithm 1 in the Supplementary Material.

**Comparison with FPL [10].** FPL studies on FL among clients with distinct domain datasets and is considered a state-of-the-art method. FPL is specifically designed to address imbalances in client distribution across domains, aiming to neutralize the skewed influence of domains with more clients on the global model training. **In contrast**, our study concentrates on the unequal learning obstacles that vary by domain, a challenge that persists regardless of equal client distribution, leading to distinct learning hurdles across domains and thereby leading to our two key operations. **Firstly**, although both our method and FPL employ prototype clustering, the objectives and implementations markedly differ. FPL's clustering is intended to harmonize the impact of local prototypes from each domain, involving only global (server-level) clustering. Conversely, our method integrates dual-level clustering at both the client (local) and server (global) levels. Our technique distinguishes itself by performing local prototype clustering, capturing critical variance and not just the mean data, which is particularly crucial in hard domains. At the global level, our clustering aims to reduce communication overhead and privacy risks by limiting the prototype variety each client sends, thereby enhancing both efficiency and privacy. **Secondly**, we introduce an innovative $\alpha$-sparsity prototype loss that features a corrective component to reasonably utilize the variance information to reduce feature similarity across different classes while boosting it within the same class, promoting more effective and stable learning.

## 5 Experiments

**Datasets.** We evaluate our proposed algorithm on three datasets: Digit-5 [42], Office-10 [8] and DomainNet [26]. **Digit-5** is a dataset for digits recognition, consisting of 5 domains: MNIST, SVHN, USPS, Synth and MNIST-M. **Office-10** is a dataset for office item recognition, consisting of 4 domains: Amazon, Caltech, DSLR and Webcam. **DomainNet** is a large-scale classification dataset, consisting of 6 domains: Clipart, Infograph, Painting, Quickdraw, Real and Sketch.

**Baselines.** We compare our algorithm with classic FL methods: FedAvg [21], FedProx [18], FedPL methods: FedProto [32], FedPCL [33], FPL[10] and FL method on feature skew: FedFA [43].

**Implement Details.** We employ the ResNet10 [9] as our backbone model, configuring the feature vectors' dimension to 512. The optimization is done using the SGD optimizer, employing a learning rate of $0.01$, momentum of $0.5$, and a weight decay of $1e-5$. For Digit-5, Office-10 and DomainNet, we use 5, 4 and 6 clients respectively. The client data is independent and identically distributed (i.i.d.) and non i.i.d. results can be found in Supplementary Material. It is important to note that each client operates within distinct data domains, meaning different datasets. For Digit-5 and Office-10, each client possessed 100 training samples and 1000 test samples. Global communication rounds are fixed at $T = 50$ for Digit-5 and $T = 80$ for Office-10. Each local training epoch consists of $E = 2$ iterations. We maintain default hyper-parameter values: $\tau = 0.07$, $\alpha = 0.25$, and $\lambda = 100$. As for DomainNet, we followed the setup in FedPCL using a 10-class subset. Each client employs 300 training samples and all test samples (approximately 1000, vary on domains). $\lambda = 1$ and $T = 200$ for DomainNet. The batch size is set at 32 for all datasets. Details regarding hyper-parameter settings will be elaborated in Sec. 5.2. For fair comparisons, we conduct each setting for 5 experiments and report the average result.

### 5.1 Performance Comparison

We compare our proposed method with the state-of-the-art methods using the Digit-5, Office-10 and DomainNet datasets, as detailed in Tab. 1, Tab. 2 and Tab. 6. Our method demonstrates significant improvements in average accuracy over baseline methods for both datasets. A closer examination of domain-specific results reveals a more pronounced enhancement in performance on domains that are more challenging to learn. For instance, within the Digit-5 dataset, our method achieves a $5.3\%$ increase in accuracy for the SVHN domain, which is more difficult, and a $0.58\%$ increase for the MNIST dataset, considered easier. This aligns with the goal of our method, which is to

Table 1: **Test accuracy on Digit-5**. Avg means average results among all clients. Details in Sec. 5.1.

| Methods | Digit-5 | | | | | | |
| --- | --- | --- | --- | --- | --- | --- | --- |
| | MNIST | SVHN | USPS | Synth | MNIST-M | Avg | Δ |
| FedAvg | $84.98 \pm 0.92$ | $29.38 \pm 1.06$ | $82.36 \pm 1.18$ | $47.00 \pm 0.73$ | $53.14 \pm 0.78$ | 59.37 | - |
| FedProx | $85.72 \pm 1.50$ | $28.86 \pm 1.23$ | $82.30 \pm 0.75$ | $46.78 \pm 1.10$ | $52.60 \pm 2.37$ | 59.25 | -0.12 |
| FedProto | $88.60 \pm 0.72$ | $31.94 \pm 1.58$ | $85.54 \pm 0.34$ | $51.82 \pm 1.12$ | $56.86 \pm 0.42$ | 62.95 | +3.58 |
| FedPCL | $88.84 \pm 1.08$ | $39.70 \pm 2.25$ | $84.74 \pm 0.72$ | $54.70 \pm 1.18$ | $59.96 \pm 1.34$ | 65.59 | +6.22 |
| FedFA | $89.46 \pm 0.55$ | $38.96 \pm 1.69$ | $85.86 \pm 0.38$ | $58.04 \pm 1.06$ | $61.38 \pm 0.98$ | 66.74 | +7.37 |
| FPL | $90.12 \pm 1.39$ | $36.78 \pm 1.88$ | $86.10 \pm 0.66$ | $57.36 \pm 1.96$ | $64.02 \pm 1.38$ | 66.88 | +7.51 |
| **Ours** | $\mathbf{90.70 \pm 0.39}$ | $\mathbf{42.08 \pm 1.59}$ | $\mathbf{86.24 \pm 1.37}$ | $\mathbf{60.08 \pm 1.47}$ | $\mathbf{67.16 \pm 0.77}$ | **69.25** | **+9.88** |

Table 2: **Test accuracy on Office-10**. Details in Sec. 5.1.

| Methods | Office-10 | | | | | |
| --- | --- | --- | --- | --- | --- | --- |
| | Amazon | Caltech | DSLR | Webcam | Avg | Δ |
| FedAvg | $48.26 \pm 1.92$ | $35.11 \pm 0.96$ | $57.29 \pm 1.47$ | $71.75 \pm 0.80$ | 53.10 | - |
| FedProx | $47.74 \pm 0.65$ | $36.44 \pm 1.92$ | $56.25 \pm 4.42$ | $73.45 \pm 0.80$ | 53.47 | +0.37 |
| FedProto | $49.31 \pm 2.18$ | $36.07 \pm 0.91$ | $57.38 \pm 2.55$ | $79.05 \pm 2.40$ | 55.45 | +2.35 |
| FedPCL | $53.65 \pm 2.33$ | $38.93 \pm 2.73$ | $58.13 \pm 5.08$ | $78.64 \pm 0.83$ | 57.34 | +4.24 |
| FedFA | $56.46 \pm 2.15$ | $40.91 \pm 2.39$ | $60.00 \pm 4.68$ | $78.58 \pm 1.86$ | 58.99 | +5.89 |
| FPL | $54.38 \pm 1.02$ | $38.24 \pm 2.38$ | $61.25 \pm 3.19$ | $80.34 \pm 1.73$ | 58.55 | +5.45 |
| **Ours** | $\mathbf{57.03 \pm 1.45}$ | $\mathbf{42.71 \pm 1.04}$ | $\mathbf{61.50 \pm 2.02}$ | $\mathbf{81.36 \pm 1.86}$ | **60.65** | **+7.55** |

address the disparate learning challenges across various domains, enhancing fairness and aiding in performance improvement in harder domains. Similar improvements can also be observed in Office-10 and DomainNet in Sec. C of Appendix.

## 5.2   Ablation Study

To evaluate the effectiveness of each component within our proposed methodology, we conducted a series of ablation studies using the Digit-5 dataset.

### 5.2.1   Impact of Dual-Level Prototype Generation.

In this subsection, we focus on examining the impact of dual-level clustered prototypes, as demonstrated by the results in Tab. 3. The first row illustrates outcomes derived from calculating local prototypes by averaging features of samples within the same class, and global prototypes through the direct averaging of these local prototypes. This approach, utilizing straightforward averaging for both levels, yields the least effective performance. This is attributed to its failure to capture variance information, a non-trivial aspect in this context. Moreover, adopting a solely global clustering approach does not significantly enhance performance. This is attributed to the fact that while global clusters incorporate inter-domain variance, they overlook the critical aspect of high intra-domain sample variance, particularly in domains that are challenging to learn. To elucidate this, we present a t-SNE visualization analysis in Fig. 3 comparing the three prototype generation methods. Our approach fosters a more generalizable decision boundary, as illustrated in the visualization. This ability to effectively capture variance at both local and global levels is key to why our dual-level clustering method outperforms the others.

Table 3: **Comparison on prototype generation methods. Variance** means the average distance from the normalized feature vector of one sample to its corresponding class feature center (i.e. the averaged prototype). Results are then used for visualization in Fig. 3. Details in Sec. 5.2.1.

| Local | Global | MNIST | SVHN | USPS | Synth | MNIST-M | Avg | Variance |
| --- | --- | --- | --- | --- | --- | --- | --- | --- |
| Avg | Avg | 86.90 | 33.10 | 83.90 | 53.40 | 61.40 | 63.54 | 1.725 |
| Avg | Cluster | 89.40 | 37.00 | 85.50 | 56.60 | 63.50 | 66.40 | 1.393 |
| **Cluster** | **Cluster** | **90.20** | **43.70** | **86.90** | **61.20** | **65.40** | **69.48** | **0.825** |

To further explore alternative methods, we examine the efficacy of directly transferring all local prototypes to each client without employing any aggregation or clustering techniques. This approach

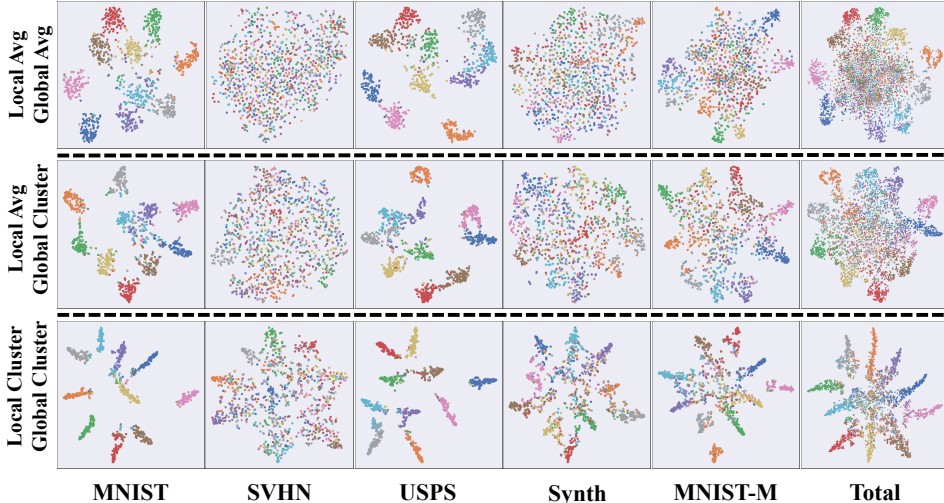

**MNIST**  **SVHN**  **USPS**  **Synth**  **MNIST-M**  **Total**

Figure 3: **Visualization of different prototype generation methods.** The **first** row averages feature vectors locally and averages local prototypes globally. The **second** row averages feature vectors locally and clusters local prototypes globally. The **last** row (**ours**) clusters feature vectors locally and clusters local clustered prototypes globally. The **last column Total** is the visualization of mixing the feature vectors from all datasets. Details in Sec. 5.2.1.

preserves variance information but raises significant privacy concerns, unlike our dual-level clustering method, which offers enhanced privacy protection. As demonstrated in Tab. 4, directly transferring all local prototypes yields performance comparable to our dual-cluster approach. However, it requires transmitting approximately five times as many prototypes in each training round.

Table 4: **Comparison between w/o and w/ global clustering. w/o** means the server distributes all local clustered prototypes to the clients for local training. **Avg # of prototypes** is the average number of prototypes each client receives from the server during each global round. Details in Sec. 5.2.1.

| Global Clustering | Avg | Avg # of Prototypes | Privacy Preservation | Communication Cost |
|---|---|---|---|---|
| w/o | $69.18 \pm 0.77$ | 100.92 | $\times$ | $4.76\times$ |
| **w/** | $\mathbf{69.47 \pm 0.71}$ | **21.20** | $\checkmark$ | $1\times$ |

### 5.2.2 Impact of $\alpha$-Sparsity Prototype Loss.

To evaluate the specific impact of our proposed $\alpha$-sparsity prototype loss, we conduct experiments comparing both contrastive and corrective loss terms. Results in Tab. 5 demonstrates that employing the contrast term led to a $1.46\%$ improvement in final average accuracy, while the correction term resulted in a $0.99\%$ enhancement. Combining all components yields the best performance, showcasing a $2.29\%$ improvement. Our investigation also focuses on the sparsity parameter $\alpha$, depicted in Fig. 4. We find that varying $\alpha$ within the range of $(0, 1)$ consistently outperforms the baseline setting with $\alpha = 1$. After careful consideration of numerical stability, we opt for an $\alpha$ value of $0.250$.

Table 5: **Comparison on components of $\alpha$-sparsity prototype loss**. Contrast and Correction stand for the contrastive and corrective loss term in the total $\alpha$-sparsity loss respectively. Avg is the average accuracy result for all clients. Details in Sec. 5.2.2.

| Contrast | Correction | Avg | $\Delta$ |
|---|---|---|---|
| w/o | w/o | $66.96 \pm 0.85$ | - |
| w/ | w/o | $68.42 \pm 0.95$ | +1.46 |
| w/o | w/ | $67.95 \pm 0.63$ | +0.99 |
| **w/** | **w/** | $\mathbf{69.25 \pm 0.62}$ | **+2.29** |

Comparing our method to the prior work FPL, which also integrates its proposed prototype loss with the CE loss, we explore different prototype loss weights, denoted by $\lambda$. The results depicted in Fig. 4 substantiate the superiority of our approach over FPL across various $\lambda$ settings.

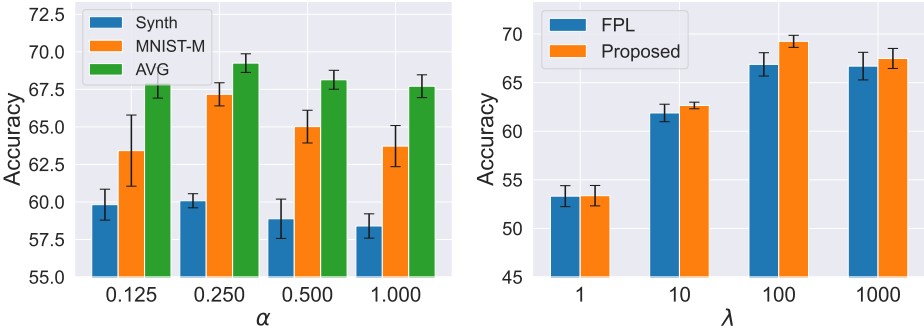

Figure 4: **Impact of $\alpha$ sparsity and $\lambda$ prototype loss weight.** The **left** figure shows the accuracy of two selected datasets and the average accuracy among all clients with different $\alpha$. The **right** figure shows the effects of different $\lambda$ for both FPL and our proposed approach. Details in Sec. 5.2.2.

### 5.3 Impact of Temperature $\tau$.

To assess the impact of the contrastive temperature ($\tau$) on our model's performance, we conduct the following experiment. The results, depicted in Fig. 5, demonstrate that our method consistently surpasses the baselines across a range of temperatures (refer to Tab. 1 and Tab. 2 for details, where FPL exhibits the highest baseline performance at $66.88\%$ and $58.55\%$ respectively). Further analysis indicates that an optimal temperature setting for our model on Digit-5 is $\tau = 0.070$. Another results on Office-10 identifies an optimal temperature setting for our model at $\tau = 0.045$, while we maintain $\tau = 0.070$ for stability in performance comparisons. It is noteworthy that temperatures either significantly higher or lower than this value lead to training difficulties due to numerical instability.

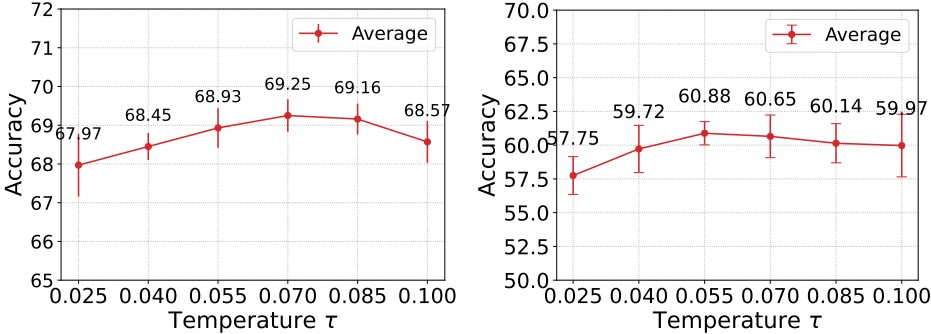

Figure 5: **Impact of $\tau$.** The **left** figure shows the impact on the Digital-5 dataset while the **right** figure shows on the Office-10 dataset. Average means the average test accuracy with variance among all clients. Details in Sec. 5.3.

## 6 Conclusion

In this paper, we start by noting the significant difference in domain-specific representation variance across various datasets within the context of federated learning involving heterogeneous data domains. Traditional methods relying on averaged prototypes, calculated as the mean values of feature representations from samples within the same class, consistently fail to capture this essential local information. Our proposed approach, FedPLVM, addresses this issue by implementing a dual-level prototype generation method. This method leverages first-level local clustering to manage variance information and employs second-level global clustering to streamline communication complexity while ensuring privacy. Additionally, we introduce an $\alpha$-sparsity prototype loss that prioritizes expanding the inter-class distance, considering the diverse cross-domain variances, and includes a correction term to subsequently reduce the intra-class distance. Comprehensive experiments have been conducted to validate the effectiveness of FedPLVM, demonstrating a significant accuracy improvement over state-of-the-art federated prototype learning methods.

## Acknowledgements

This work is supported in part by NSF under grants 2033681, 2006630, 2044991, 2319780.

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

# A   Supplementary Material Overview

The supplementary material is organized into the following sections:

**B:** Detailed algorithm of our proposed method, FedPLVM.

**C:** Further results of our experiments.

**D:** Additional quantitative results comparing FedPLVM and other baseline methods under the label non-i.i.d. setting.

**E:** Additional quantitative results comparing different clustering algorithms for the prototype clustering procedure of FedPLVM.

**F:** Additional quantitative results comparing FedPLVM and FPL[10] under the unbalanced client domain distribution setting.

**G:** Additional quantitative results employing differential privacy technology in conjunction with FedPLVM.

**H:** Additional quantitative results comparing different global prototype generation methodologies of FedPLVM.

**I:** Additional quantitative results analyzing the effectiveness of local prototypes clustering on different domains.

**J:** Discussion on the wider range of scenario of our method, including limitation and broader impact.

# B   Detailed Algorithm of FedPLVM

We summarize our proposed method, FedPLVM in the Alg. 1.

---

**Algorithm 1** FedPLVM

---

**Input**: Communication rounds $T$, local training epochs $E$, number of classes $M$, number of clients $K$, private dataset $\mathcal{D}_k = \{\mathbf{x}_i, y_i\}_i^{N_k}$
**Output**: Global model $w^{T+1}$
**Server Aggregation**:
1: **for** $t = 1, 2, ..., T$ **do**
2:     **for** $k = 1, 2, ..., K$ **do**
3:         Collect local models and clustered prototypes $w_{k,E+1}^t, \{\mathcal{P}_k^m\}_{m=1}^M \leftarrow$ **Local Update** $(k, w^t, \mathcal{G})$
4:     **end for**
5:     Aggregate collected prototypes $\{\mathcal{P}^m\}_{m=1}^M$
6:     Generate global clustered prototypes $\mathcal{G}$ in Eq. 4
7:     Aggregate global model $w^{t+1} = \sum_{k=1}^K \frac{N_k}{N} w_{k,E+1}^t$
8: **end for**
**Local Update**$(k, w^t, \mathcal{G})$:
1: $w_{k,1}^t \leftarrow w^t$
2: **for** $e = 1, 2, ..., E$ **do**
3:     Update $w_{k,e+1}^t$ from $w_{k,e}^t$ using $\mathcal{G}$ by Eq. 10
4: **end for**
5: Compute local feature vectors $\{h(\mathbf{x}_i)\}_{i=1}^{N_k}$ for $\mathcal{D}_k$
6: Generate local clustered prototypes $\{\mathcal{P}_k^m\}_{m=1}^M$ in Eq. 3
7: Return $w_{k,E+1}^t, \{\mathcal{P}_k^m\}_{m=1}^M$

---

# C   Further results on DomainNet

Due to page limitation, we put some further experimental results in Sec. 5.1 here:

Table 6: **Test accuracy on DomainNet.** Details in Sec. 5.1.

| Methods | DomainNet | | | | | | Avg | Δ |
|---|---|---|---|---|---|---|---|---|
| | Clipart | Infograph | Painting | Quickdraw | Real | Sketch | | |
| FedAvg | 50.62 ± 0.58 | 27.31 ± 0.91 | 42.71 ± 2.72 | 14.11 ± 1.26 | 43.71 ± 3.39 | 34.67 ± 2.80 | 35.52 | - |
| FedProx | 52.33 ± 1.25 | 28.22 ± 0.11 | 42.56 ± 3.32 | 13.50 ± 0.94 | 45.85 ± 1.19 | 35.46 ± 2.98 | 36.32 | +0.80 |
| FedProto | 54.08 ± 2.11 | 30.44 ± 1.70 | 47.20 ± 3.34 | 19.40 ± 3.36 | 50.57 ± 2.07 | 44.22 ± 2.61 | 40.99 | +5.47 |
| FedPCL | 53.37 ± 1.76 | 29.53 ± 3.08 | 46.69 ± 3.51 | 16.32 ± 1.22 | 51.36 ± 1.85 | 43.32 ± 0.12 | 40.10 | +4.58 |
| FedFA | 53.20 ± 0.50 | 29.22 ± 1.50 | 47.12 ± 1.42 | 17.90 ± 0.65 | 50.14 ± 1.37 | 45.76 ± 1.61 | 40.56 | +5.04 |
| FPL | 53.13 ± 0.65 | 27.55 ± 1.86 | 45.40 ± 2.29 | 17.47 ± 1.36 | 50.64 ± 1.33 | 47.64 ± 2.45 | 40.31 | +4.79 |
| **Ours** | **54.19 ± 1.73** | **31.14 ± 1.68** | **47.22 ± 1.10** | **22.40 ± 1.91** | **51.66 ± 1.73** | **48.70 ± 2.03** | **42.55** | **+7.03** |

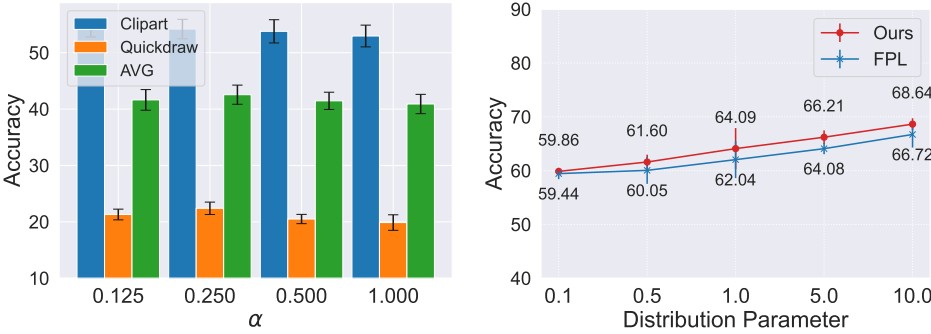

Figure 6: **Impact of $\alpha$ on the DomainNet dataset.**

Figure 7: **Test accuracy on DomainNet with different Dirichlet distribution parameters.**

## D   Label Non-I.I.D. Setting

We expanded our method's assessment on Digit-5 and Office-10 within a non-i.i.d. label setting. Utilizing the Dirichlet method ($\alpha = 0.5$), we shaped the data distribution and generated non-i.i.d. datasets for individual clients. Notably, our approach showcased substantial accuracy improvements compared to baseline methods across both datasets. Non-i.i.d. distributions are visualized in Fig. 8, the variation in features across non-identically distributed data is reflected by the colors of the dots, whereas the non-identical label distribution ($\alpha = 0.5$) is denoted by the sizes of the dots. For the top figure in Fig. 8, client index $0, 1, 2, 3, 4$ own the datasets MNIST, SVHN, USPS, Synth and MNIST-M respectively; for the bottom figure, client index $0, 1, 2, 3$ represent the dataset domains Amazon, Caltech, DSLR and Webcam repectively.

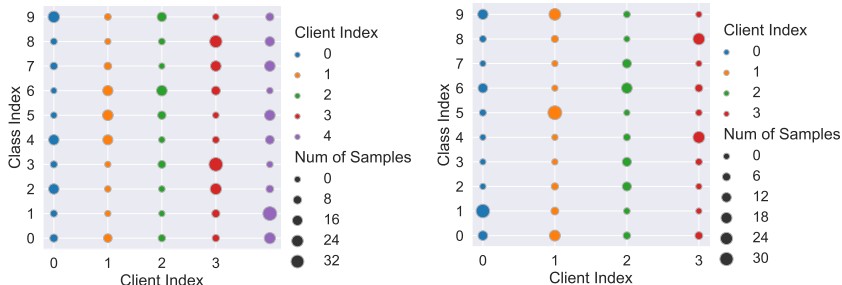

Figure 8: **Visualization of data distributions under label non-i.i.d. setting**. Every point on the graph symbolizes a collection of samples belonging to a specific class assigned to a client. The **left** figure is from digit-5 and the **right** figure is from office-10. Details in Sec. D

The outcomes are presented in Tab. 7 and Tab. 8. Consistent with our findings in the i.i.d. label setting, we observed a significant performance boost in domains that pose greater learning challenges. For instance, within the Digit-5 dataset, our method achieved a $2.24\%$ accuracy surge in the Synth domain—a more complex task—while showing a $0.94\%$ increase in accuracy for the comparatively

easier USPS dataset. This reiterates our method's aim: addressing disparate learning difficulties among domains, fostering fairness, and bolstering performance, especially in tougher domains. Similar advancements were evident within the Office-10 dataset.

Table 7: **Test accuracy on Digit-5 under label non-i.i.d. setting**. Avg means average results among all clients. We apply the Dirichlet method ($\alpha = 0.5$) to obtain the data distribution and create the non-i.i.d. dataset for each client. Details in Sec. D.

| Methods | Digit-5 | | | | | | |
| --- | --- | --- | --- | --- | --- | --- | --- |
| | MNIST | SVHN | USPS | Synth | MNIST-M | Avg | $\Delta$ |
| FedAvg | $79.44 \pm 0.86$ | $24.16 \pm 0.91$ | $65.46 \pm 1.86$ | $40.58 \pm 0.87$ | $55.16 \pm 1.24$ | 52.96 | - |
| FedProx | $81.40 \pm 0.53$ | $24.44 \pm 1.06$ | $67.54 \pm 2.05$ | $41.64 \pm 1.02$ | $55.80 \pm 0.58$ | 54.16 | +1.20 |
| FedProto | $83.28 \pm 0.80$ | $23.06 \pm 0.51$ | $69.98 \pm 0.81$ | $40.78 \pm 0.99$ | $57.66 \pm 0.85$ | 54.95 | +1.99 |
| FedPCL | $85.44 \pm 2.13$ | $25.82 \pm 0.97$ | $69.28 \pm 3.73$ | $42.60 \pm 1.55$ | $58.66 \pm 1.46$ | 56.36 | +3.40 |
| FedFA | $86.65 \pm 0.69$ | $29.57 \pm 1.97$ | $73.48 \pm 0.77$ | $47.19 \pm 0.49$ | $60.24 \pm 1.24$ | 59.43 | +6.47 |
| FPL | $85.18 \pm 2.66$ | $31.54 \pm 2.96$ | $72.96 \pm 2.41$ | $49.48 \pm 2.42$ | $61.08 \pm 2.10$ | 60.05 | +7.09 |
| **Ours** | $\mathbf{86.18 \pm 0.98}$ | $\mathbf{33.24 \pm 1.31}$ | $\mathbf{73.98 \pm 2.17}$ | $\mathbf{51.72 \pm 1.08}$ | $\mathbf{62.90 \pm 1.32}$ | **61.60** | **+8.64** |

Table 8: **Test accuracy on Office-10 under label non-i.i.d. setting**. Avg means average results among all clients. We apply the Dirichlet method ($\alpha = 0.5$) to obtain the data distribution and create the non-i.i.d. dataset for each client. Details in Sec. D.

| Methods | Office-10 | | | | | |
| --- | --- | --- | --- | --- | --- | --- |
| | Amazon | Caltech | DSLR | Webcam | Avg | $\Delta$ |
| FedAvg | $47.36 \pm 1.60$ | $33.63 \pm 0.21$ | $49.42 \pm 1.37$ | $72.95 \pm 1.38$ | 50.84 | - |
| FedProx | $48.58 \pm 0.76$ | $33.78 \pm 0.63$ | $52.42 \pm 5.31$ | $75.10 \pm 0.80$ | 52.47 | +1.63 |
| FedProto | $52.44 \pm 0.86$ | $35.56 \pm 1.09$ | $51.38 \pm 0.19$ | $75.62 \pm 2.11$ | 53.75 | +2.91 |
| FedPCL | $54.07 \pm 0.76$ | $36.56 \pm 2.18$ | $57.62 \pm 0.11$ | $76.42 \pm 0.82$ | 56.17 | +5.33 |
| FedFA | $53.28 \pm 1.72$ | $36.67 \pm 1.49$ | $53.71 \pm 1.43$ | $73.20 \pm 2.52$ | 54.22 | +3.38 |
| FPL | $53.05 \pm 0.50$ | $35.41 \pm 1.05$ | $58.67 \pm 2.95$ | $76.62 \pm 1.60$ | 55.94 | +5.10 |
| **Ours** | $\mathbf{55.28 \pm 0.76}$ | $\mathbf{37.48 \pm 0.21}$ | $\mathbf{59.79 \pm 1.90}$ | $\mathbf{77.36 \pm 0.08}$ | **57.48** | **+6.64** |

# E  Different Clustering Algorithms

In our paper, FINCH is employed for our dual-level prototype clustering due to its parameter-free nature. Here we select the one with the minimum number of cluster centers among several possible clustering schemes generated by FINCH. This characteristic obviates the need for hyper-parameter tuning typically required in clustering methods, such as selecting the number of centers in K-Means algorithm. However, this does not imply exclusivity to the FINCH clustering algorithm within our method. Our approach is compatible with various clustering algorithms, including simpler methods like K-Means. The experiments result in Tab. 9 show that with a carefully tuned K-Means algorithm (precisely determining the appropriate number of centers), our method with K-Means achieves performance comparable to that achieved using FINCH. Conversely, poorly tuned parameters result in weaker performance with the K-Means clustering. This reinforces our choice of FINCH, for its parameter-free advantage.

Table 9: **Comparison with K-Means Algorithm. Adaptive K** means we use the number of clustering centers from FINCH as K. Details in Sec. E.

| Local | Global | MNIST | SVHN | USPS | Synth | MNIST-M | Avg |
| --- | --- | --- | --- | --- | --- | --- | --- |
| Adaptive K | Adaptive K | 89.97 | 42.10 | 85.83 | 60.90 | 66.03 | 68.97 |
| K = 2 | K = 2 | 89.53 | 40.67 | 85.33 | 59.13 | 63.53 | 67.64 |
| K = 5 | K = 5 | 89.72 | 41.97 | 85.41 | 60.13 | 64.69 | 68.38 |
| **FINCH** | **FINCH** | **90.70** | **42.08** | **86.24** | **60.08** | **67.16** | **69.25** |

# F  Unbalanced Clients Distribution Setting

We further explore the performance of our FedPLVM in handling unbalanced client distributions, mirroring the scenario demonstrated in FPL. Within a pool of 10 clients, the data ownership is

distributed as follows: 1 client possesses MNIST domain data, 4 clients hold SVHN domain data, 2 clients possess USPS domain data, 2 clients hold Synth domain data, and 1 client owns MNIST-M domain data.

The outcomes, presented in Tab. 10, showcase our approach's equitable performance, notably in enhancing test accuracy on clients dealing with the more challenging datasets. Comparatively, in contrast to FPL, our method displays a significant $2.94\%$ improvement in SVHN dataset performance. It is worth noting that our 'avg' column in Tab. 10 represents the average test accuracy across all clients, differing from FPL, which calculates the average test accuracy first among clients with the same dataset and then averages across datasets. This distinction addresses potential unfairness, wherein a higher representation of clients with 'easy' datasets could disproportionately influence the final accuracy in FPL.

Table 10: **Comparison on unbalanced clients distribution.** Test accuracy on each dataset domain is the average result among all clients that own the corresponding dataset. Avg means average results among all clients. Details in Sec. F.

| Methods | MNIST | SVHN | USPS | Synth | MNIST-M | Avg |
|---|---|---|---|---|---|---|
| FPL | $90.44 \pm 0.67$ | $59.78 \pm 1.56$ | $85.14 \pm 0.93$ | $73.01 \pm 1.09$ | $69.34 \pm 0.98$ | 71.52 |
| **Ours** | $\mathbf{90.90 \pm 0.97}$ | $\mathbf{62.72 \pm 1.69}$ | $\mathbf{85.92 \pm 0.75}$ | $\mathbf{74.51 \pm 1.20}$ | $\mathbf{71.32 \pm 0.35}$ | **73.40** |

# G  Privacy Protection

In our method, the server only receives clustered local prototypes, making it challenging to reconstruct the clients' local datasets. We also employ a differential privacy (DP) technology to validate the impact of FedPLVM. Each client trains local model by DP-SGD [1] to perturb model parameters. The noise multiplier is determined by [22, 4, 3]. The privacy budget $\epsilon$ and approximate parameter $\delta$ are set as $4.0$ and $1e - 5$ respectively for a $(\epsilon, \delta)$-DP setting. Meanwhile, we also incorporate the clustered local prototypes with the privacy protection technique. We set the scale parameter $s = 0.05$ and the perturbation coefficient $p = 0.1$ for the Gaussian noise distribution generation of our local clustered prototypes. As shown in Tab. 11, the average accuracy drops at most $0.77\%$ if we employ the privacy protection only for the prototypes. The approximate DP causes a slightly larger decrease in accuracy, up to $2.45\%$. Note that even with the privacy protection technologies, our method reaches a comparable performance to the most advanced baseline in Tab. 1.

Table 11: **Impact of differential privacy.** Avg means average results among all clients. **w/** and **w/o** represents we incorporate the local model or the local clustered prototypes with the privacy protection technologies or not. Details in Sec. G.

| Model | Prototypes | MNIST | SVHN | USPS | Synth | MNIST-M | Avg |
|---|---|---|---|---|---|---|---|
| w/ | w/ | $89.74 \pm 0.35$ | $38.40 \pm 2.86$ | $84.44 \pm 0.48$ | $55.98 \pm 1.27$ | $62.44 \pm 1.25$ | 66.20 |
| w/ | w/o | $89.42 \pm 0.85$ | $38.12 \pm 1.12$ | $83.68 \pm 0.43$ | $57.54 \pm 1.74$ | $65.26 \pm 0.51$ | 66.80 |
| w/o | w/ | $90.64 \pm 0.34$ | $40.12 \pm 1.95$ | $86.40 \pm 0.99$ | $58.80 \pm 1.53$ | $66.44 \pm 1.60$ | 68.48 |
| w/o | w/o | $\mathbf{90.70 \pm 0.39}$ | $\mathbf{42.08 \pm 1.59}$ | $\mathbf{86.24 \pm 1.37}$ | $\mathbf{60.08 \pm 1.47}$ | $\mathbf{67.16 \pm 0.77}$ | **69.25** |

# H  Global Prototypes Generation

The determination of the number of global prototypes for a class is contingent on the associated learning challenges. In instances where a class within a domain presents significant learning difficulties, it is characterized by large variance. Consequently, employing a single global prototype, reducing the sufficiency of variance information, could decay the learning performance. Incorporating multiple global prototypes for such a class can introduce additional variance information, thereby enhancing the learning process. We provide averaged experiment results in Tab. 12. (Note in the main paper we only report the single experiment to be consistent with the corresponding visualization.) The results validate the necessity of our dual-level clustering.

Table 12: **Comparison with Local Cluster & Global Average.** Details in Sec. H.

| Local | Global | MNIST | SVHN | USPS | Synth | MNIST-M | Avg |
|--------|---------|-----------------|-----------------|-----------------|-----------------|-----------------|-------|
| Cluster | Avg | $89.43 \pm 0.55$ | $40.57 \pm 1.21$ | $85.74 \pm 0.89$ | $59.77 \pm 0.88$ | $65.03 \pm 1.02$ | 68.11 |
| **Cluster** | **Cluster** | $\mathbf{90.70 \pm 0.39}$ | $\mathbf{42.08 \pm 1.59}$ | $\mathbf{86.24 \pm 1.37}$ | $\mathbf{60.08 \pm 1.47}$ | $\mathbf{67.16 \pm 0.77}$ | **69.25** |

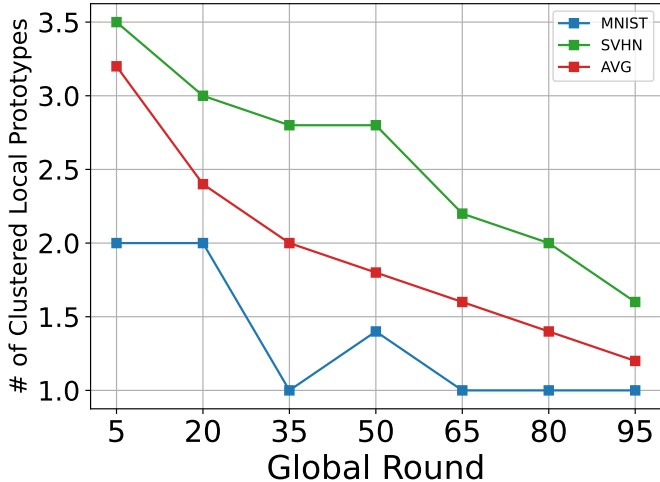

Figure 9: **Tendency of average number of local clustered prototypes for different classes in different domains.** Details in Sec. I.

# I Local Prototypes Clustering

Local prototype clustering is beneficial because it captures essential variance information, not just the average data, which is particularly crucial in hard domains. Easy domains tend to show tight clustering within the same category and clear distinctions between different categories, facilitating accurate classification. In contrast, hard domains often exhibit looser clustering, increasing the risk of misclassification, especially for samples near category boundaries. Therefore, only capturing average data suffices for easy domains but falls short for hard domains.

Our proposed method captures more feature distribution variation information by introducing the local prototypes clustering, since clustering provides more prototypes compared to simply averaging especially considering the sparse distribution in hard domains as shown in Fig. 3. For easy domains, where the average is sufficient, our method generates fewer prototypes.

To demonstrate this, we refer to Fig. 9. The y-axis shows the average number of local prototypes among classes generated at each selected round. The easy domain (MNIST) has fewer prototypes generated compared to the hard domain (SVHN), showing that in hard domains, more prototypes are utilized to better capture the variance information. Furthermore, an average performance gain of 3.08% in experiment of impact of local prototypes clustering in Tab. 3 also supports this observation.

# J Discussion on Wider Range of Scenarios

## J.1 Limitation.

While FedPLVM introduces innovative strategies to address domain heterogeneity in federated learning, it also comes with certain limitations and challenges. For example, noisy labels, or incorrectly labeled data instances, are a common issue in real-world datasets. A few previous works has discussed the related impact in FL scenario [37, 38, 36]. This noisy case will bring two problems for prototype learning in FL: **Prototype Distortion.** Noisy labels can lead to distorted prototype representations, as the local prototypes clustered for each class may be influenced by mislabeled instances. This distortion can propagate throughout the training process, affecting the model's ability to accurately capture the true underlying data distribution. **Prototype Ambiguity.** In the presence

of noisy labels, the distinction between different classes becomes less clear, leading to ambiguity in prototype definitions. Prototypes may no longer accurately represent the true characteristics of their respective classes, making it difficult for the model to generalize effectively. These make noisy labels in FL scenario an interesting direction for future extension to our proposed FedPLVM.

### J.2 Broader Impact.

FedPLVM primarily addresses the challenges of cross-domain federated learning, where data from heterogeneous domains need to be collaboratively utilized while preserving privacy. This capability enables the application of FedPLVM in various real-life scenarios, including: **Healthcare Data Collaboration.** Different hospitals or healthcare institutions often possess diverse patient datasets with varying characteristics and distributions. FedPLVM allows these institutions to collaboratively train machine learning models for tasks such as disease diagnosis, patient outcome prediction, or personalized treatment recommendation while ensuring data privacy and security. **Internet of Things (IoT) Networks.** IoT devices deployed in different locations or environments generate heterogeneous data streams, including sensor readings, environmental data, and user interactions. FedPLVM enables federated learning across IoT networks, facilitating tasks such as anomaly detection, predictive maintenance, or environmental monitoring without centralized data collection, thus preserving user privacy and reducing communication overhead.

