# OpenReview forum: "Taming Cross-Domain Representation Variance in Federated Prototype Learning with Heterogeneous Data Domains"
_NeurIPS.cc/2024/Conference — NeurIPS 2024 poster_

### Official Review · Reviewer_gr7B · 2024-06-14

**Soundness:** 3
**Presentation:** 3
**Contribution:** 2
**Rating:** 6
**Confidence:** 5

**Summary:**

This paper studies federated learning under heterogeneous data domains and introduces a federated prototype learning strategy, denoted as FedPLVM, to mitigate the problem. A dual-level prototype generation method is proposed to address domain variance between hard and easy domains, reducing the communication burden between clients and the server. Moreover, an \alpha-sparsity prototype loss is proposed to enhance the sparsity of the prototype representations. The experiment results demonstrate the efficacy of the proposed method and the necessity of each proposed module.

**Strengths:**

1. The paper is overall well-written and easy to follow.
2. The authors explain why models exhibit varying performance across domains.
3. The experiment results show the superiority of the proposed method compared to SOTA methods. The ablation results also indicate the effectiveness of each proposed module.

**Weaknesses:**

1. The proposed dual-level clustering strategy appears similar to FPL. What is the key difference between them? The primary distinction lies in the proposed \alpha-sparsity loss, which introduces incremental innovation to the paper.
2. The proposed dual-level prototype strategy utilizes FINCH for clustering. However, the clustering results from FINCH can vary at different steps. Which step's result is selected for use? Do results from different steps vary?
3. The impact of \alpha is only evaluated on small-scale datasets. Does the trend consist with big-scale datasets like DomainNet?
4. In Sec. G, the authors claim that the evaluation protocol used in FPL is unfair. However, the protocol used in FPL is sensitive to both easy and hard datasets.

**Questions:**

This paper is well-organized and the experiment results significantly surpass existing methods. But the level of innovation is questionable: the dual-level prototype generation is similar to FPL, and the only key difference is the \alpha-sparsity prototype loss. Moreover, I also have some concerns on this paper. For more details, please check the weakness part.

**Limitations:**

Yes. The authors addressed limitations of the paper.

---

> ### Author Rebuttal · Authors · 2024-08-07
>
> Thank you for your insightful review. Here is our response to the mentioned weaknesses:
>
> 1. The key point of our dual-level clustering is local prototypes clustering. Different from easy domains, hard domains often exhibit looser clustering, increasing the risk of misclassification, especially for samples near category boundaries. Our dual-level prototypes clustering captures more feature distribution variation information by introducing the local prototypes clustering operation, since clustering locally provides more prototypes compared to simply averaging especially considering the sparse distribution in hard domains as shown in Figure 3 of our paper.
>
>     To demonstrate this, we refer to Figure 1 of the attached pdf file. The y-axis shows the average number of local prototypes among classes generated at each selected round. Note that the number of prototypes for different classes can vary, resulting in non-integer averages. The easy domain (MNIST) has fewer prototypes generated compared to the hard domain (SVHN), showing that in hard domains, more prototypes are utilized to better capture the variance information. Furthermore, an average performance gain of 3.08%  in experiment of impact of local prototypes clustering in Table 2 of our paper also supports this observation. At the global level, our clustering aims to reduce communication overhead and privacy risks by limiting the prototype variety each client sends, enhancing both efficiency and privacy while FPL intends to balance the impact of local prototypes from each domain, involving only server-level clustering.
>
>     Meanwhile, our proposed α-sparsity prototype loss is also innovated and designed to complement the dual-level clustered prototypes operation. This loss function origins from reducing the potential risk of feature representation overlapping caused by our dual-level clustered prototypes operation. This loss function reasonably utilizes the variance information to reduce feature similarity across different classes while increasing it within the same class, promoting more effective and stable learning.
>
> 2. For each clustering operation, the installed FINCH python project provides several clustering results and we select the one with the smallest number of clustering prototypes. During different rounds, the number of local clustered prototypes vary for different classes among different clients, generally decreasing as training progresses. From Figure 1 of the attached pdf file, we observe that the average number of clustered local prototypes consistently decreases over the training process. This indicates that our local prototypes clustering method aligns with the convergence process among different domains.
>
> 3. We further conduct the ablation study of \alpha on the DomainNet dataset as shown in Figure 2 of the attaced pdf file. Our extensive experiments indicate that maintaining $\alpha$ within the stable range of 0.125 to 0.25 consistently yields the best performance. Although the results vary with different values of $\alpha$, our method consistently outperforms all baseline methods, demonstrating its robustness.
>
> 4. In FPL, the client domain distribution is unbalanced, with data ownership distributed among 20 clients as follows: MNIST (3 clients), USPS (7 clients), SVHN (6 clients), and SYN (4 clients). However, the reported final average accuracy is computed by summing the results from all clients and dividing by 20, which gives more weight to domains with more clients, leading to an unfair representation.
>
>     In Section G, we address this issue by first averaging the results within each domain based on the number of clients in that domain. We then average these domain-specific results based on the number of domains. This weighted averaging ensures that all domains are given equal weight, regardless of the number of clients they have.
>
> For Question 1, please refer to our response to Weakness 1.

---

> > ### Comment · Reviewer_gr7B · 2024-08-11
> >
> > Thank you for your response, some of my concerns are solved. However, I still do not find a distinction between dual-cluster and FPL. And for R4, the weighted average could be another valuable evaluation protocol but I do not think the protocol used in FPL and other methods is unfair. Thus, I would like to maintain my score.

---

> > > ### Author Response · Authors · 2024-08-11
> > >
> > > Thank you for your continued feedback. We appreciate the opportunity to further clarify our approach.
> > >
> > > 1. We would like to further clarify the distinctions between our method and FPL. Our dual-level prototype clustering significantly differs from the single-level global prototype clustering  FPL, both in purpose and effectiveness, as we detailed in our previous response and demonstrated in the visualization results of our paper. The local prototype clustering is crucial because it captures essential variance information, rather than just representing the average feature as in FPL. This is particularly important in challenging domains. For instance, as shown in Figure 1 of the attached PDF, the easier domain (MNIST) generates noticeably fewer prototypes compared to the harder domain (SVHN), which underscores the necessity of our local prototype clustering approach. Moreover, our proposed $\alpha$-sparsity loss introduces an innovative and effective enhancement to our dual-level prototype clustering. We conducted additional experiments to highlight the advantages of the  $\alpha$--sparsity loss in terms of both convergence speed and final accuracy. In the Digit-5 experiment, we performed 100 global rounds as stated in our paper and compared the convergence speed with and without the $\alpha$-sparsity loss. The results, which report the average test accuracy after every 10 rounds, clearly show that the $\alpha$--sparsity loss leads to faster convergence (achieving it in 70 rounds compared to 80 rounds without the proposed loss) and higher final accuracy.
> > > | Rounds            |  10   |  20   |  30   |  40   |  50   |  60   |  70   |  80   |  90   | 100   |
> > > |:-----------------:|:-----:|:-----:|:-----:|:-----:|:-----:|:-----:|:-----:|:-----:|:-----:|:-----:|
> > > | w/o \(\alpha\)-sparsity loss | 35.14 | 56.44 | 57.58 | 62.20 | 64.90 | 65.72 | 65.02 | 66.66 | 66.32 | 66.06 |
> > > | w/ \(\alpha\)-sparsity loss  | 54.12 | 62.44 | 65.32 | 66.12 | 67.32 | 68.20 | 69.62 | 69.22 | 69.86 | 69.26 |
> > >
> > > 2. We appreciate your insightful comment. Following your suggestion, we have calculated the additional average accuracy using the protocol in FPL. Our updated results show that the average accuracy of our proposed method is 77.07%, compared to 75.54% for FPL. This further demonstrates the superior performance of our method.

---

> > > > ### Comment · Reviewer_gr7B · 2024-08-12
> > > >
> > > > Thank you for your detailed response and addressed my concerns. Thus, I would like to raise my score.

---

### Official Review · Reviewer_8nBN · 2024-06-15

**Soundness:** 3
**Presentation:** 2
**Contribution:** 2
**Rating:** 4
**Confidence:** 5

**Summary:**

This paper focuses on Federated Prototypes Learning and reveals that existing methods create the same prototypes for different clients, which neglects the distribution diversity. In this work, the authors introduce the variance-aware dual-level prototype clustering and alpha sparsity prototype loss. Various experiments demonstrate the effectiveness of the proposed method.

**Strengths:**

Authors conduct comprehensive experiments to demonstrate the effectiveness.

**Weaknesses:**

This paper has several drawbacks.

1. Motivation for the α-Sparsity Prototype Loss. It is a little strange that "This multiplicity of prototypes could potentially lead to overlapping feature representations among different classes, especially in challenging client scenarios". As the authors claim the feature overlapping, the naive solution is to use other ways to cover the local feature behavior rather than multiple local prototypes.  Furthermore, I assume that authors could employ the OT (optimal transport) to require them to concentrate on different parts. I do not think the current operation is a suitable way to deal with the feature overlapping.

2. The paper architecture is not suitable. I wonder why authors spend a lot of space to write meaningless or complicated words and even leave no space for DomainNet and Office-10. Furthermore, I encourage to consider the label skew effect, ie, with different label skew degree. Besides, the experiments on large scale of clients are also important.

3. The Figure 1 is confusing. Why monitor would capture the digitis figure? It is a not soundable problem figure.

**Questions:**

I encourage the author to take careful thinking for the federated prototype learning field. The existing solution shares a high similarity with the FPL (CVPR'23)  and seems like an incremental work.

**Limitations:**

Yes, the authors have discussed.

---

> ### Author Rebuttal · Authors · 2024-08-07
>
> Thank you for your insightful review. Here is our response to the mentioned weaknesses:
>
> 1. We introduce \alpha-sparsity loss to mitigate the potential risk of feature representation overlapping caused by the dual-level clustered prototypes operation. This loss focuses on maximizing inter-class distance and putting more attention on expanding the overall feature distribution to a broader range, thereby solving the feature overlapping problem efficiently and simply. While Optimal Transport (OT) may be employed in our framework, it introduces additional significant computational complexity [1]. Given our experimental results, it is effective to use our proposed \alpha-sparsity loss due to its simplicity.
>
>     [1]. Peyré G, Cuturi M. Computational optimal transport: With applications to data science[J]. Foundations and Trends® in Machine Learning, 2019, 11(5-6): 355-607.
>
> 2. We apologize for any confusion caused by our paper's structure. Could you please provide us with a range of 'meaningless' and 'complicated' words so we can revise them accordingly? Moreover, we can move the DomainNet and Officie-10 parts from the Appendix to the main text per your suggestion in our final version.
>
>     Regarding the label skew effect, we conduct label non-IID experiments using the Dirichlet method with a distribution parameter of 0.5, as detailed in Section E. The label distribution becomes more non-IID as the distribution parameter decreases. To illustrate the impact of different distribution parameters (0.1, 0.5, 1, 5, and 10), we compare the average accuracy with the best baseline method FPL from our previous non-IID experiment on Digit-5. Figure 3 of the attached pdf file shows that our method consistently outperforms FPL across all distribution parameters. However, the performance gap narrows as the datasets become more non-IID. This trend is expected, as in non-IID datasets, the number of samples for some classes can be very small, leading to less representative feature representations. Consequently, our clustered local prototypes may become similar to the averaged single prototype used in FPL.
>
>     We also conduct the experiments on a larger scale of 20 clients, the same number in FPL, and the datasets of per four clients come from the same domain. As shown in Table 1 of the attached pdf file, our method still outperforms FPL under such a setting. Note we have already explored the unbalanced clients distribution setting of 10 clients in Section G.
>
> Question 1:
>
>   Firstly, while both our method and FPL employ prototype clustering, the objectives and implementations are markedly different. FPL performs single-level global clustering, intended to balance the impact of local prototypes from each domain, involving only server-level clustering.  This single-level global clustering is specifically designed to address imbalances in client distribution across domains, aiming to neutralize the skewed influence of domains with more clients on global model training. Conversely, our method integrates dual-level clustering at both the client (local) and server (global) levels. At the global level, our clustering aims to reduce communication overhead and privacy risks by limiting the prototype variety each client sends, enhancing both efficiency and privacy. Our local prototype clustering is crucial because it captures essential variance information, not just the average data, which is particularly crucial in hard domains. hard domains often exhibit looser clustering, increasing the risk of misclassification, especially for samples near category boundaries. Therefore, only capturing average local prototype suffices for easy domains but falls short for hard domains. To demonstrate this, we refer to Figure 1 of the attached pdf file. The y-axis shows the average number of local prototypes among classes generated at each selected round. The easy domain (MNIST) has fewer prototypes generated compared to the hard domain (SVHN), showing that in hard domains, more prototypes are utilized to better capture the variance information. Furthermore, an average performance gain of 3.08%  in experiment of impact of local prototypes clustering in Table 2 of our paper also supports this observation.
>
>   Secondly, we introduce an innovative $\alpha$-sparsity prototype loss featuring a corrective component, which mitigates the potential risk of feature representation overlapping caused by the dual-level clustered prototypes operation.. This loss function reasonably utilizes the variance information to reduce feature similarity across different classes while increasing it within the same class, promoting more effective and stable learning.

---

> ### Comment · Reviewer_8nBN · 2024-08-09
>
> Thank you for the AC tips. I have provided more details in my response to the review.
>
> As for the details of the review,
> 1. The naive solution is to utilize a Gaussian distribution to describe class information. For example, FedFA [1] is based on Gaussian modeling of feature statistic augmentation. I encourage the authors to compare the conceptual differences between local prototype clustering and class Gaussian construction. Additionally, the authors should clearly explain the advantages of utilizing a dual-level prototype approach.
> 2. With respect to paper organization, the authors should spend more space on concept comparison with existing methods. But the authors focus on preliminary discussion.
>
>
> Thank you for authors feedback. I still have the following questions:
> 1. You refer to a paper from 2019, but many papers related to Optimal Transport (OT) have been published in recent years. For instance, FedOTP [2] introduces the OT loss in regularization.
> 2. The authors did not address question 3: “Figure 1 is confusing. Why would a monitor capture the digits? It does not seem like a logical problem illustration.”
> 3. You mentioned that “This loss function reasonably utilizes the variance information to reduce feature similarity across different classes while increasing it within the same class, promoting more effective and stable learning.” Could the authors provide the corresponding theoretical or convergence analysis? Additionally, regarding “which mitigates the potential risk of feature representation overlapping,” could the authors offer visualizations to support this claim rather than relying on terms like “potential”?
>
> [1] FedFA: Federated Feature Augmentation
> [2] Global and Local Prompts Cooperation via Optimal Transport for Federated Learning. CVPR 2024

---

> > ### Author Response · Authors · 2024-08-11
> >
> > Thank you for your further comments.
> >
> > For the details of the review:
> > 1. FedFA aims to augment the feature statistic by constructing a vicinity Gaussian distribution for each data point and change the empirical risk to the vicinal risk. Such an augmentation operation can enrich the feature information after the feature extractor layer. However, this approach has its limitations, especially in cases where data volume is small or distribution is uneven. In these scenarios, the constructed Gaussian distribution might not fully capture the true diversity of the data and can have bias. Also, it requires additional computation regarding the distribution parameters locally for each feature extractor layer with augmentation.  In contrast, our proposed FedPLVM utilizes only the feature representation of the final feature extractor layer to cluster several local prototypes regardless of the even or uneven feature distribution and further performs global clustering and combines with the $\alpha$-sparsity loss in the local training. Lastly, FedFa is one of the baselines we compared in our paper and the result shows our method outperforms FedFA in Digit-5, Office-10 and DomainNet.
> >
> > 2. We appreciate your suggestions regarding the organization of our paper. However, we must respectfully disagree with the comment that excessive space is devoted to preliminary discussions at the expense of comparisons with existing methods. Firstly, the preliminary section occupies only half a page and concisely presents essential knowledge required for understanding our proposed method. This information is crucial and cannot be further condensed without loss of clarity. Secondly, we have integrated comparisons with existing methods throughout the introduction and methodology sections. Specifically, our comparison with FPL, which is most relevant to our work, includes dedicated paragraphs discussing the differences (see lines 232-247).
> >
> > For Questions:
> >
> > 1. We appreciate for suggesting the paper on FedOTP. While FedOTP is indeed a noteworthy contribution, we believe it addresses a different aspect from that of our paper. FedOTP focuses on developing efficient collaborative prompt learning strategies to integrate pre-trained visual-language models into federated learning frameworks. This topic, although valuable, is not closely related to the core focus of our research.
> >
> > 2. The icon in Figure 1 of our paper mentioned in your question is not that of a monitor but a laptop. In FL, the client device doesn’t necessarily need to capture the data itself. In our illustration, the laptop processes the stored digital data without needing to capture figures or photos directly.
> >
> > 3. We thank you for your advice on including a convergence speed analysis. We agree that theoretical convergence analysis is a crucial aspect of federated learning. Typically, such analyses in existing FL research depend heavily on strong assumptions about gradient bounds and the differences between global and local loss functions. Importantly, incorporating prototype learning alters the loss function itself. Since FL convergence analyses require the specific assumptions regarding the loss functions, the advantages of integrating prototype learning into the loss function might not be readily apparent. This limitation is also why existing FL studies that utilize prototype learning, such as FPL, do not provide theoretical convergence analysis. Instead, to address concerns regarding convergence speed, we have provided experimental results that clearly demonstrate the improvements our method achieves. In our Digit-5 experiment, we performed 100 global rounds as stated in our paper and compared the convergence speed with and without our proposed $\alpha$-sparsity loss. We reported the average test accuracy after each 10 rounds training. Obviously with the $\alpha$-sparsity loss we have a faster convergence speed (70 rounds compared to 80 rounds without the proposed loss function) and a higher converged accuracy:
> > | Round | 10 | 20 | 30 | 40 | 50 | 60 | 70 | 80 | 90 | 100 |
> > | :---: | :---: | :---: | :---: | :---: | :---: | :---: | :---: | :---: | :---: | :---: |
> > | w/o $\alpha$-sparsity loss | 35.14 | 56.44 | 57.58 | 62.20 | 64.90 | 65.72 | 65.02 | 66.66 | 66.32 | 66.06 |
> > | w/ $\alpha$-sparsity loss | 54.12 | 62.44 | 65.32 | 66.12 | 67.32 | 68.20 | 69.62 | 69.22 | 69.86 | 69.26 |
> >
> > 4. We appreciate your suggestion to include visualizations to support our claims. However, during the author-reviewer discussion period, we are unable to add new figures. Should our paper be accepted, we will certainly include the recommended visualizations in the final version of the manuscript as advised.

---

> > > ### Comment · Reviewer_8nBN · 2024-08-12
> > >
> > > Thanks for your response. I maintain my score.

---

### Official Review · Reviewer_dkgn · 2024-07-08

**Soundness:** 3
**Presentation:** 4
**Contribution:** 3
**Rating:** 6
**Confidence:** 3

**Summary:**

This paper aims to investigate the federated prototype learning problem with data heterogeneity. To handle the cross-domain representation variance problem, a new method termed FedPLM is proposed, which includes a dual-level prototype clustering mechanism and an alpha-sparsity prototype loss. Experiments are conducted to show the effectiveness of the proposed method.

**Strengths:**

1. The proposed method is well-designed for the specific research problem and has good novelty.

2. The motivation of the paper is solid.

3. The experiments are comprehensive to discuss the properties of the proposed method.

4. The paper is well-written and easy to follow.

**Weaknesses:**

1. On several datasets (e.g. MNIST and USPS), the performance of the proposed method is not significant enough. Can the author give more explanation for this point?

2. The running efficiency of the proposed method is not given. Complexity analysis and efficiency-related experiments are expected to discuss the efficiency of the proposed method.

**Questions:**

The authors are expected to address the concerns in the block of "Weaknesses". Also, I have one more question: How to balance the two sub-terms (contra and corr) in alpha loss? As they may be on different scales, is that feasible to directly add them together?

**Limitations:**

Limitations are discussed in the paper. For the negative societal impact, I didn't find any concern from my side.

---

> ### Author Rebuttal · Authors · 2024-08-07
>
> Thank you for your insightful review. Here is our response to the mentioned weaknesses:
>
> 1. Our design principle aims to ensure our method excels particularly in challenging domains, which often have lower baseline accuracy, such as SVHN. This aligns with our motivation to tackle diverse learning challenges across various domains. In more challenging domains, the dual-level clustering method generates multiple clustered local prototypes that capture variance information effectively, thus enhancing focus and fairness, as shown in Figure 1 of the attached pdf file. For easier datasets like MNIST, where feature representations naturally concentrate, our local clustering tends to produce a centralized prototype similar to the traditional average method, leading to a smaller performance gap.
>
>     We focus on hard domains beacuse in real-world settings, difficult domains are common and pose significant challenges that need effective solutions. Previous works often struggle in these hard domains, and our method addresses this gap by ensuring robust performance improvements where they are most needed.
>
>     Contrary to the concern about insignificant results, our method does show considerable improvements in challenging domains. For instance, on the SVHN, Synth and MNIST-M, our approach achieved a notable performance gain of 5.3%, 2.72% and 3.14% respectively compared to SOTA methods. This demonstrates the effectiveness and robustness of our method in handling difficult datasets, which are critical for practical applications.
>
> 2. For the computation cost of FedPLVM, the feature representations of data samples can be obtained directly from the model without any additional computation. Our method introduces the dual-level prototypes clustering compared to other baselines, and we test the average running time for local training (0.0296s), local prototypes clustering (0.552s) and global prototypes clustering (0.0214s). Compared to the local training, our additional dual-level clustering step is negligible (approximately 0.0296 / 0.552 ≈ 5% of the local training time). Meanwhile, the proposed \alpha-sparsity loss does not change the network structure.
>
>     For the communication cost of FedPLVM, we need each client to upload local clustered prototypes with the local model to the server and the client will then download the global clustered prototypes with the global model for the local training of the next round. Our modified lightweight ResNet-10 has approximately 4.9 million parameters. In our network, one prototype is only a 512-dimension vector, so the maximum number of total uploaded parameters for the local clustered prototypes can be approximately 3.2 (from the highest point of the red line in the above picture ) * 10 (number of classes) * 512 = 0.016 million and the estimated value of the downloaded parameters for the global clustered prototypes can be 21.20 (from table 3 in the paper) * 512 = 0.011 million which are both ignorable compared to the size of the model. Note the previous work FedPCL requires each client download the prototypes from all other clients from the server, which makes our clustering prototypes operation more realizable.
>
> Question 1: A straightforward approach to balance the two sub-terms is introducing another hyper-parameter to control the weight. However, to maintain the simplicity of our proposed \alpha loss, we can adjust the temperature $\tau$ in the contrastive term to balance the two sub-terms. In our experiments, using our default $\tau$, the average loss value of the first term in the 50th round out of 100 is 0.364, while the second term is 0.105, demonstrating comparable values.

---

### Official Review · Reviewer_GDmh · 2024-07-11

**Soundness:** 3
**Presentation:** 3
**Contribution:** 3
**Rating:** 5
**Confidence:** 4

**Summary:**

The paper introduces FedPLVM, a federated learning approach that improves federated prototype learning (FedPL) in heterogeneous data domain setting. Traditional FedPL methods create the same number of prototypes for each client, leading to performance disparities across clients with different data distributions. FedPLVM mitigates this by introducing variance-aware dual-level prototype clustering and an α-sparsity prototype loss. Specifically, each client first clusters the feature vectors of all same-class local samples to get local clustered prototypes. Then,  the server conducts global prototype clustering to reduce the number of prototypes for each class. Experiments show that FedPLVM outperforms the other baselines on the data from multiple domains.

**Strengths:**

1. The motivation is clear. It makes sense to get different prototypes under tasks with different difficulties.

2. The proposed \alpha-sparsity prototype loss is interesting. The ablation study about the loss and dual-level prototype generation is clear.

**Weaknesses:**

1. The paper misses important baselines. Personalized federated learning studies are not studied, which should be suitable in the studied setting.

2. The computation and communication cost of FedPLVM compared with other studies is not presented.

3. The clustering method needs more details. It is not clear why clustering can address the heterogeneous domain issue.

4. The work is incremental work based on FedPL.

**Questions:**

1. From Algorithm 1, the goal of FedPLVM is to train a single global model. From the experiments, the model is tested on the local datasets. Given data from different domains, why don’t the parties train personalized local models with personalized federated learning?

2. The paper mentioned that more prototypes are needed for harder domains. For the clustering method, is there a guarantee that the harder domain will also get more local prototypes? The paper may add more analysis or experiments to demonstrate it.

3. Will FedPL have significant higher computation and communication cost than FedAVg?

**Limitations:**

Yes.

---

> ### Author Rebuttal · Authors · 2024-08-07
>
> Thank you for your insightful review. Here is our response to the mentioned weaknesses:
>
> 1. Firstly, we want to highlight that the main objective of this paper is to develop a global model that works across all domain datasets. This approach aligns with the baseline methods, such as FedProto and FPL, which also address variations in client data domains.
>
>     Secondly, training a unified global model, as opposed to personalized models, offers distinct advantages. For instance, a potential use case involves applying the trained model to broader scenarios, such as when a client has data from multiple domains and the domain of each data sample is unknown. In such cases, a global model remains effective, whereas personalized models may not be suitable.
>
>     Lastly, developing a strong global model enhances our ability to implement personalization effectively. While our method can integrate personalization techniques, it is not the current focus of our work. We plan to explore personalization in more depth in our future research.
>
> 2. For the computation cost of FedPLVM, the feature representations of data samples can be obtained directly from the model without any additional computation. Our method introduces the dual-level prototypes clustering compared to other baselines, and we test the average running time for local training (0.0296s), local prototypes clustering (0.552s) and global prototypes clustering (0.0214s). The additional local prototypes clustering in dual-level clustering step is negligible, accounting for only about 0.0296 / 0.552 ≈ 5% of the local training time. Meanwhile, the proposed \alpha-sparsity loss does not change the network structure.
>
>     For the communication cost of FedPLVM, each client uploads local clustered prototypes along with the local model to the server. The client then downloads the global clustered prototypes and the global model for the next round of local training. Our employed ResNet-10 has approximately 4.9 million parameters. Each prototype is a 512-dimension vector, so the maximum number of uploaded parameters for the local clustered prototypes is approximately 3.2 (from the highest average number of local clustered prototypes of the red line in Figure 1 of the attached pdf file) * 10 (number of classes) * 512 = 0.016 million and the estimated value of the downloaded parameters for the global clustered prototypes can be 21.20 (from table 3 in the paper) * 512 = 0.011 million.  Both are negligible compared to the model size. Notably, the previous work FedPCL requires each client to download the prototypes from all other clients via the server, making our clustering prototypes operation more efficient and feasible.
>
> 3. Local prototype clustering is beneficial because it captures essential variance information, not just the average data, which is particularly crucial in hard domains. Easy domains tend to show tight clustering within the same category and clear distinctions between different categories, facilitating accurate classification. In contrast, hard domains often exhibit looser clustering, increasing the risk of misclassification, especially for samples near category boundaries. Therefore, only capturing average data suffices for easy domains but falls short for hard domains.
>
>     To demonstrate this, we refer to Figure 1 of the attached pdf file. The y-axis shows the average number of local prototypes among classes generated at each selected round. The easy domain (MNIST) has fewer prototypes generated compared to the hard domain (SVHN), showing that in hard domains, more prototypes are utilized to better capture the variance information. Furthermore, an average performance gain of 3.08%  in experiment of impact of local prototypes clustering in Table 2 of our paper also supports this observation.
>
> 4. We respectfully disagree. There are a few key distinctions between our work and FedPL.
>
>     Firstly, while both our method and FPL employ prototype clustering, the objectives and implementations are markedly different. FPL performs single-level global clustering, intended to balance the impact of local prototypes from each domain, involving only server-level clustering.  This single-level global clustering is specifically designed to address imbalances in client distribution across domains, aiming to neutralize the skewed influence of domains with more clients on global model training. Conversely, our method integrates dual-level clustering at both the client (local) and server (global) levels. At the global level, our clustering aims to reduce communication overhead and privacy risks by limiting the prototype variety each client sends, enhancing both efficiency and privacy. Our local prototype clustering is crucial because it captures essential variance information, not just the average data, which is particularly crucial in hard domains. Hard domains often exhibit looser clustering, increasing the risk of misclassification, especially for samples near category boundaries. Therefore, only capturing average local prototype suffices for easy domains but falls short for hard domains.
>
>     Secondly, we introduce an innovative α-sparsity prototype loss, highlighted as “interesting” in the strengths part of your review, featuring a corrective component, which mitigates the potential risk of feature representation overlapping caused by the dual-level clustered prototypes operation. This loss function reasonably utilizes the variance information to reduce feature similarity across different classes while increasing it within the same class, promoting more effective and stable learning.
>
> For Question 1, 2 and 3, please refer to our response to Weakness 1, 3 and 2 respectively.

---

> > ### Comment · Reviewer_GDmh · 2024-08-10
> >
> > Thanks for the author's response. Some of my concerns have been addressed. I'll increase my score to 5.

---

### Official Review · Reviewer_e95T · 2024-07-12

**Soundness:** 3
**Presentation:** 3
**Contribution:** 3
**Rating:** 5
**Confidence:** 5

**Summary:**

The domain gap among multiple clients impedes the generalization of federated learning. To mitigate cross-domain feature representation variance, the authors introduce FedPLVM, which establishes variance-aware dual-level prototypes clustering and employs a novel α-sparsity prototype loss. To verify the effectiveness of the proposed method, extensive experiments are conducted.

**Strengths:**

1. This paper is well organized and written in a way that is easy to understand.
2. The experimental design is reasonable and a large number of experiments have also proved the effectiveness of the proposed methodology.

**Weaknesses:**

1. It makes sense to perform global level prototype clustering on the server side, while performing intra-class prototype clustering on the local side makes unclear sense, why it is advantageous to get more prototypes for hard domain samples?
2. In Eq. (8), what is meaning of C_y?
3. From Fig. 4, \alpha has a significant impact on the results, which is not conducive to the robustness of the algorithm.
4. The latest compared method in the paper is from 2023 and lacks the latest comparison method.

**Questions:**

Please refer to the weaknesses.

**Limitations:**

Please refer to the weaknesses.

---

> ### Author Rebuttal · Authors · 2024-08-07
>
> Thank you for your insightful review. Here is our response to the mentioned weaknesses:
> 1.  Local prototype clustering is beneficial because it captures essential variance information, not just the average data, which is particularly crucial in hard domains. Easy domains tend to show tight clustering within the same category and clear distinctions between different categories, facilitating accurate classification. In contrast, hard domains often exhibit looser clustering, increasing the risk of misclassification, especially for samples near category boundaries. Therefore, only capturing average data suffices for easy domains but falls short for hard domains.
>
>     Our proposed method captures more feature distribution variation information by introducing the local prototypes clustering, since clustering provides more prototypes compared to simply averaging especially considering the sparse distribution in hard domains as shown in Figure 3 of our paper. For easy domains, where the average is sufficient, our method generates fewer prototypes.
>
>     To demonstrate this, we refer to Figure 1 of the attached pdf file. The y-axis shows the average number of local prototypes among classes generated at each selected round. The easy domain (MNIST) has fewer prototypes generated compared to the hard domain (SVHN), showing that in hard domains, more prototypes are utilized to better capture the variance information. Furthermore, an average performance gain of 3.08%  in experiment of impact of local prototypes clustering in Table 2 of our paper also supports this observation.
>
> 2. $C_y$ is the number of clustered prototypes for the class $y$.
>
> 3. Firstly, our extensive experiments indicate that maintaining $\alpha$ within the stable range of 0.125 to 0.25 consistently yields the best performance. We further conduct the ablation study on the large-scale Domain dataset. As shown in Figure 2 of the attached pdf file, we can still observe a performance benefit from the stable range.
>
>     Secondly, it's important to note that $\alpha$ is a crucial element in our proposed $\alpha$-sparsity loss. If $\alpha$ is set to 1, our $\alpha$-sparsity loss reduces to a standard contrastive loss, although it still includes a specially designed corrective term (and also the proposed dual-level prototypes clustering operation). Therefore, it's natural for $\alpha$ to impact the results.
>
>     Lastly, in our ablation study (refer to Figure 4 in the main paper), although the results vary with $\alpha$, our method consistently outperforms all baseline methods, demonstrating its robustness.
>
> 4. To the best of our knowledge, there is still no newer work under the same setting. If you are aware of any such studies, please let us know, and we will compare our work with them. Furthermore, there is one recent fair federated learning work, FedHeal [1], which compares with other fair federated learning methods by evaluating whether they can work well with some baseline methods used in our work, such as FedAVG and FedProto and bring any benefits. Although it operates under different setting compared to ours and does not compare with any baseline works in our setting, we conducted experiments to demonstrate that our proposed framework performs better than the SOTA method FPL + FedHeal in [1]. From Table 1 of the attached pdf file, we can observe that while FPL gains a performance boost from FedHeal, our proposed method still outperforms it.
>
>     [1]. Chen Y, Huang W, Ye M. Fair Federated Learning under Domain Skew with Local Consistency and Domain Diversity[C]//Proceedings of the IEEE/CVF Conference on Computer Vision and Pattern Recognition. 2024: 12077-12086.

---

> > ### Comment · Reviewer_e95T · 2024-08-12
> >
> > Thank you for your responses. In light of the novelty and other reviewers' comments, I tend to maintain my score.

---

### Author Rebuttal · Authors · 2024-08-07

We are grateful to all reviewers for their insightful feedback and recognition of our work. We particularly appreciate Reviewer GDmh’s comment on the well-designed of our proposed $\alpha$-sparsity prototype loss, Reviewer dkgn's comment on the novelty of our proposed dual-level prototypes clustering method and the other reviewers’ affirmation that our experiments are comprehensive and validate our proposed method effectively. We also value the feedback pointing out areas of weakness and the questions raised about our paper. We prepare the separate respective response to each reviewer about their specific concerns. Below, we first provide a consolidated response addressing the main concerns raised by multiple reviewers:

1. Benefit from Local Prototype Clustering:

    Local prototype clustering effectively captures essential variance information, not just averages, which is vital in complex domains. Our method generates more prototypes for challenging domains to better capture this variability, while fewer prototypes are sufficient in simpler domains where averages provide enough detail. This strategy is depicted in Figure 1 of the attached pdf file, which shows fewer prototypes in the easier domain (MNIST) and more in the harder domain (SVHN), highlighting the necessity for more prototypes in complex scenarios to adequately capture variance.

2. Computation and Communication Costs:
     - Computation: Our approach introduces dual-level prototypes clustering, adding a minimal overhead—only about 5% of the total local training time—compared to other baselines.
     - Communication: Each client uploads only the extra local clustered prototypes to the server. Given the size of the large local model (4.9 million parameters), the additional prototypes (approximately 0.01 million parameters) are negligible.

3. Distinction from FPL:

    Our objectives and implementations differ significantly from FPL. FPL employs single-level global clustering, aiming to harmonize local prototypes’ impact across domains, involving only server-level clustering. In contrast, our approach addresses the varying learning challenges across domains, utilizing dual-level clustering at both the client (local) and server (global) levels. At the global level, our clustering strategy reduces communication overhead and privacy risks by limiting the variety of prototypes each client sends, thus enhancing both efficiency and privacy. Our local prototype clustering is crucial in complex domains, where it captures essential variance information, critical for reducing misclassification risks near category boundaries. This is not necessary in easier domains where capturing average data suffices.

    Additionally, we introduce the innovative $\alpha$-sparsity prototype loss with a corrective component that mitigates the risk of feature representation overlap caused by dual-level clustered prototypes. This loss function effectively utilizes variance information to decrease feature similarity across different classes while increasing it within the same class, promoting more effective and stable learning.

---

### Author Response · Authors · 2024-08-11

Dear Reviewers,

We sincerely appreciate your extensive efforts in reviewing and commenting on our work. We hope that our rebuttal has effectively addressed your comments and concerns.

As the author-reviewer discussion period is approaching to its end on August 13th, we kindly encourage you to reach out if you have any further questions or points of discussion. Your feedback is highly valued, and we are eager to address any additional remaining concerns you may have.

Thank you very much for your attention to this matter.

Anonymous Authors

---

### Decision · Program_Chairs · 2024-09-25

**Decision:**

Accept (poster)

**Comment:**

The paper presents a variance-aware dual-level prototype clustering and a $\alpha$-sparsity prototype loss to tackle performance loss on hard domains during federated prototype learning. A majority of the reviewers believe that the paper is very well written and addresses an important shortcoming of the prior work federated prototype learning.

The authors present strong experimental results to support their methodology and provide appropriate intuitions and visualizations to further improve the readability of the work. The additional results presented by the authors in the rebuttal resolved most of the concerns that were raised by the reviewers regarding (i) space and time complexity, (ii) differences with respect to the prior works, (iii) differences in performance gap among baselines on different datasets, and (iv) details of the FINCH clustering.

I would request the authors to consider these forum discussions and appropriately reorganize the paper before final submission. I believe it would improve the overall quality of the work and help the readers understand the method better.